# Experimental observation of localized interfacial phonon modes

Zhe Cheng [1,10,11], Ruiyang Li[2,11], Xingxu Yan [3,4,11], Glenn Jernigan[5], Jingjing Shi [1], Michael E. Liao [6], Nicholas J. Hines [1], Chaitanya A. Gadre[7], Juan Carlos Idrobo [8], Eungkyu Lee [9], Karl D. Hobart[5], Mark S. Goorsky[6], Xiaoqing Pan [3,4,7 ✉], Tengfei Luo [2 ✉] & Samuel Graham [1 ✉]

Interfaces impede heat flow in micro/nanostructured systems. Conventional theories for interfacial thermal transport were derived based on bulk phonon properties of the materials making up the interface without explicitly considering the atomistic interfacial details, which are found critical to correctly describing thermal boundary conductance. Recent theoretical studies predicted the existence of localized phonon modes at the interface which can play an important role in understanding interfacial thermal transport. However, experimental validation is still lacking. Through a combination of Raman spectroscopy and high-energy-resolution electron energy-loss spectroscopy in a scanning transmission electron microscope, we report the experimental observation of localized interfacial phonon modes at ~12 THz at a high-quality epitaxial Si-Ge interface. These modes are further confirmed using molecular dynamics simulations with a high-fidelity neural network interatomic potential, which also yield thermal boundary conductance agreeing well with that measured in time-domain thermoreflectance experiments. Simulations find that the interfacial phonon modes have an obvious contribution to the total thermal boundary conductance. Our findings significantly contribute to the understanding of interfacial thermal transport physics and have impact on engineering thermal boundary conductance at interfaces in applications such as electronics thermal management and thermoelectric energy conversion.

[1] George W. Woodruff School of Mechanical Engineering, Georgia Institute of Technology, Atlanta, GA 30332, USA. [2] Department of Aerospace and Mechanical Engineering, University of Notre Dame, Notre Dame, IN 46556, USA. [3] Department of Materials Science and Engineering, University of California, Irvine, CA 92697, USA. [4] Irvine Materials Research Institute, University of California, Irvine, CA 92697, USA. [5] U.S. Naval Research Laboratory, 4555 Overlook Avenue SW, Washington, DC 20375, USA. [6] Materials Science and Engineering, University of California, Los Angeles, Los Angeles, CA 90095, USA. [7] Department of Physics and Astronomy, University of California, Irvine, CA 92617, USA. [8] Center for Nanophase Materials Sciences, Oak Ridge National Laboratory, Oak Ridge, TN 37831, USA. [9] Department of Electronic Engineering, Kyung Hee University, Yongin-si, Gyeonggi-do 17104, South Korea. [10]Present address: Department of Materials Science and Engineering, University of Illinois at Urbana-Champaign, Urbana, IL 61801, USA. [11]These authors contributed equally: Zhe Cheng, Ruiyang Li, Xingxu Yan. ✉email: xiaoqinp@uci.edu; tluo@nd.edu; sgraham@gatech.edu

nterfaces impede heat flow in micro/nanostructured systems, which highlights the growing importance of thermal management of microelectronics and devices for energy conversion and storage[1,2]. Thermal boundary conductance (TBC, $G$), the property characterizing thermal transport across an interface, is defined by a temperature drop ($\Delta T$) across an interface for a given heat flux ($J$) as $G = J/\Delta T$[3,4]. For insulator or semiconductor-related interfaces, lattice vibrations (phonons) are the dominant heat carriers for interfacial thermal transport. At the microscale, thermal transport across an interface is usually described by the phonon gas model as phonons impinging the interface, which results in a portion of the energy passing through the interface as characterized by a transmission coefficient. TBC is thus often described as $G = \left(\sum_j \int v_\omega c_\omega t_\omega d\omega_j\right)/4$, where $v_\omega, c_\omega, t_\omega$ are, respectively, group velocity, heat capacity, and transmission coefficient of the incident phonon with frequency $\omega$, and polarization $j$[4]. A critical task in the interfacial thermal transport research has been calculating the transmission coefficient[3,5]. The acoustic mismatch model (AMM) and diffuse mismatch model (DMM) are traditional models used for this purpose, and they calculate the transmission coefficient based on the phonon properties of the bulk materials making up the interface[6–8]. However, such treatments are problematic, as they ignore any microscopic details of the interface. It has been proven that factors such as bond strength, interface roughness, and atomic mixing can significantly impact TBC[1,9–12]. Atomistic Green's function (AGF) is able to explicitly include realistic atomistic structures in the TBC calculation, which can yield better predictions compared to DMM or AMM[13]. However, phonon transmission coefficients calculated using AGF are still expressed as a function of phonons of the bulk materials[11,14,15].

In recent years, several molecular dynamics (MD) studies[16–19] have shown that interfacial phonon modes can play important roles in interfacial thermal transport. These modes are mainly localized at the interfacial region due to the special interatomic bonding environment not seen in bulk materials. However, these MD simulations either were based on empirical interatomic potentials that were not specifically developed for simulating interfacial thermal transport or employed approximations for interfacial interactions despite the use of first-principles-calculated force constants[16–19]. It remains unclear whether such interfacial phonon modes indeed exist at realistic interfaces, which calls for the experimental observation of such modes. The univocal confirmation of the existence of interfacial phonon modes is going to be critical to correctly understanding interfacial thermal transport physics and potentially engineering TBC. However, to date, experiments have been impeded by the technical difficulties related to probing localized modes at relevant size scales and having a properly chosen interface that is free of other factors (e.g., phonon polariton in polar materials, roughness, etc.)[5,20–22].

In this work, we confirm the existence of interfacial phonon modes by growing high-quality Si-Ge interfaces using molecular beam epitaxy (MBE) and detecting the localized modes with Raman spectroscopy and high-energy-resolution electron energy-loss spectroscopy (EELS) in a scanning transmission electron microscope (STEM). As both Si and Ge are nonpolar materials, there are no strong delocalized phonon polariton modes in the acquired EELS signal, which enables the spatial resolution of the characterization to be atomic scale[20]. We further confirm that the detected localized modes from these experiments are indeed interfacial phonon modes using MD simulations with a high-fidelity neural network potential (NNP) trained by first-principles calculations specifically for the Si-Ge interface. The calculated

TBC values from MD agree well with the experimentally measured values using time-domain thermoreflectance (TDTR). Finally, spectral analysis in MD shows that the interfacial phonon modes can obviously contribute to the overall TBC, despite its limited population and localized nature.

## Results

To fabricate interfaces that are clean and sharp for interfacial phonon mode detection, we chose the relatively low lattice-mismatched Si-Ge interface and grew high-quality interfaces using MBE, which enabled a precise control of the sample structure. Two sample architectures were grown, each fabricated for interfacial phonon mode detection and TDTR measurements, respectively. The detailed growth processes can be found in the "Methods" section and Supplementary Information (SI). Sample 1 consists of a 10 nm Si cap on a 300 nm Ge layer grown on a Si substrate and it is used for Raman (Fig. 1a) and EELS measurements (Fig. 2a). Measurement details are included in the "Methods" section and SI. Such a structure of Sample 1 is chosen, as the absorption depth of Si is large (774 nm), tens of times greater than that of Ge, at the Raman laser wavelength (488 nm)[23]. Thus, the 10 nm Si cap layer can allow the Raman laser to transmit through to reach the Si-Ge interfaces. Sample 2 is a 250 nm Ge layer grown on a Si substrate and is used for TBC measurements, where the Ge layer thickness is optimized to maximize the sensitivity of the TBC of the Si-Ge interface in TDTR measurements (Supplementary Fig. 12). To confirm the quality of the Si-Ge interface, high-angle annular dark-field STEM (HAADF-STEM) imaging technique is used to characterize the interface of Sample 1. In Fig. 1a, the Z-contrast HAADF-STEM image shows that the interface is sharp despite a thin atomic mixing layer (Supplementary Figs. 9 and 10). The atomic structures of the two Si-Ge interfaces in Sample 1 and Sample 2 are similar (Supplementary Fig. 9).

As a comparison to the Raman spectrum of Sample 1 which involves a Si-Ge interface, we have also measured the Raman spectra of a pure Ge wafer and a pure Si wafer. Figure 1b shows that the Raman peak of bulk Ge is at ~9 THz and that of Si is at ~15.6 THz. The Si peak in Sample 1 is redshifted by 0.15 THz compared to pure Si, because the epitaxial Si cap layer is slightly strained on the Ge wafer to accommodate the ~4% lattice mismatch. Other than these two peaks originated from Ge and Si, an additional peak around 11.3–12.2 THz shows up in Sample 1, which may be attributed to the Si-Ge interface.

To further confirm that this peak originates from the interface, we employ high-energy resolution (routinely ~1.7–1.9 THz with an exposure time of 1 s) EELS, in a STEM, with a probe size of 1.5 Å to spatially resolve the phonon signal around the Si-Ge interface (see details in the "Methods" section and SI). Figure 2a shows the schematic diagram of the EELS measurements. The peaks in the EELS signal are the energies of vibrational phonon modes in the sample that inelastically interact with the incident electron probe. Figure 2b shows a line-scan of the vibrational spectra across the interface as illustrated by the arrows in Fig. 2a. Under our experimental setup with a large convergence semi-angle (33 mrad), the as-acquired vibrational spectra contain the momentum-integrated vibrational modes throughout the entire Brillouin zone (BZ) and are comparable with the phonon density of states (PDOS)[24].

Three representative EELS vibrational spectra from Si (red), Ge (green), and the interface (blue) taken at the locations marked in Fig. 2a are shown in Fig. 2c. The vibrational spectrum of the interfacial region includes the peaks from Si, Ge, and the interfacial vibration modes. Figure 2d shows the intensities of vibrational spectra at 11.6, 12.0, and 12.4 THz as a function of distance

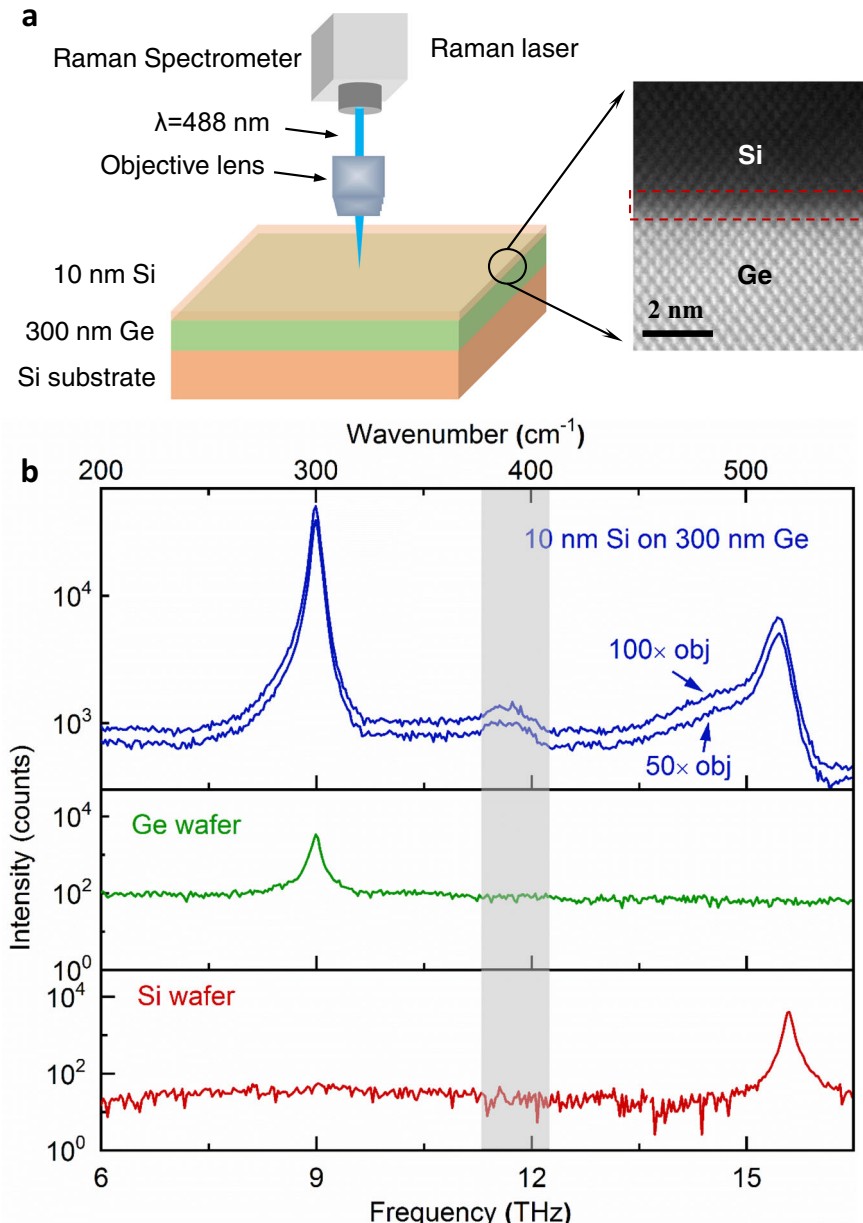

**Fig. 1 Raman detection of interfacial modes at the Si-Ge interface. a** Schematic diagram of Raman measurements on Sample 1 and its HAADF-STEM image showing a high-quality Si-Ge interface. **b** Raman spectra of Sample 1, a Ge wafer, and a Si wafer. Two objectives (×50 and ×100) were used to perform the Raman measurements on Sample 1 and the measurement details can be found in the Methods section. A distinct Raman peak around 11.3–12.2 THz from Sample 1 is not seen in bulk Ge or Si wafers, which is attributed to the interfacial modes.

to the interface. A clear peak of the vibrational spectral intensity is observed around the interface at 11.6 THz and there is a weak peak at 12.0 THz. It is notable that there are some similar intensities in the same energy range on the Si side, which arise from the bulk phonon modes in the Si as shown in Fig. 2c. Those are from the optical and acoustic phonon modes near the BZ boundary, according to the phonon dispersion curve. Two results are used to distinguish the interfacial modes from these bulk Si modes. First, the interfacial vibrational spectrum has extra intensity near 12.0 THz (Supplementary Fig. 4), compared to the linear combination of bulk Ge and Si vibrational spectra, indicating the existence of additional phonon modes at the interface. Second, we further performed angle-resolved vibrational EELS to collect the vibrational signals at the BZ center (Γ point, Supplementary Fig. 11a) to exclude the interference of phonon modes near the BZ boundary. We observed a clear peak around 12 THz

at the interface while that from the bulk Si at this frequency disappeared. By combining the Raman and high-energy-resolution EELS results above, we can claim that the detected modes around 12 THz are indeed interfacial phonon modes localized at the interface. Leveraging the high spatial resolution (1.5 Å) capability of STEM-EELS, this interfacial vibrational mode is determined to be confined within ~1.2 nm of the inter-face (Fig. 2d).

To further understand the interfacial modes, we use MD simulations to model the Si-Ge interface with the level of inter-facial mixing (~0.7 nm thick) similar to those observed in the HAADF intensity (Fig. 3a and Supplementary Fig. 9). In order to reproduce the interfacial modes, a high-fidelity NNP for the Si-Ge interface is specifically developed by training against first-principles density functional theory calculations (see "Methods" section and SI for details). The NNP-MD scheme has been

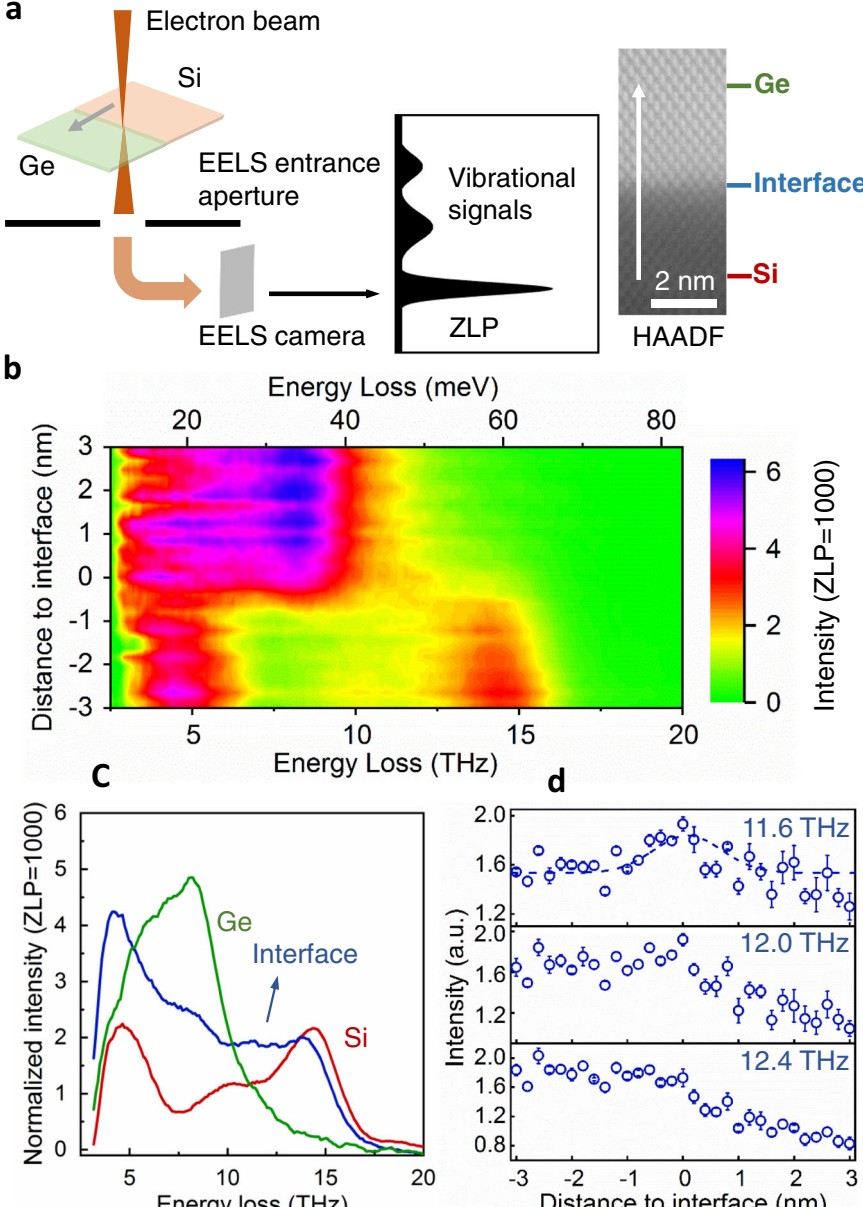

**Fig. 2 Electron energy-loss spectroscopy (EELS) in a scanning transmission electron microscope (STEM) spatially resolves the interfacial phonon modes. a** Schematic diagram of EELS measurements of vibrational modes around the Si-Ge interface and the path of a line-scan. ZLP is zero-loss peak and HAADF is high-angle annular dark field. **b** The line profile of the vibrational spectra across the interface. **c** Three representative vibrational spectra with a total acquisition time of 200 s from Si (red), Ge (green), and interface (blue) detected from EELS, as marked in **a**. **d** The integrated vibrational signal at 11.6, 12.0, and 12.4 THz in a window with width of 0.48 THz. The measured signal at 11.6 THz overlaps with a Gaussian fitting curve with a full width half maximum (FWHM) of 1.2 nm (less than two unit cells). The error bars are SD.

proved to be capable of simulating semiconductor materials and predicting thermal properties with accuracy comparable to first-principles calculations[25,26]. Using MD simulations with this NNP, PDOS at different locations throughout the simulation domain along the direction perpendicular to the Si-Ge interface is calculated (see "Methods" section and SI for details).

As shown in Fig. 3b, each panel is the PDOS of the atoms at a certain location shown in Fig. 3a. The interfacial region is composed of the mixed region and the first layer away from the mixture on each side. For the regions that are 10 nm away from the interface, the PDOS of both Si and Ge are converged to the ones of bulk materials and are not affected by the interface. The PDOS peaks of the Si and Ge optical phonons are ~15 THz and ~9 THz, which are consistent with the Raman peaks. Near the interface, a peak at ~11.4–12.8 THz emerges for both the Si and Ge atoms, which corresponds well with the interfacial modes detected from the Raman and EELS measurements. This calculated interfacial mode peak agrees better with our experiments compared to other results of MD simulations using empirical potentials. Chalopin et al.[16] obtained such a peak at ~13.5 THz using a Stinger-Weber potential, while Gordiz et al.[17] found the peak between 12–13 THz using a Tersoff potential. Our calculated peak is similar to that calculated from first-principles force constants (~12 THz), which, however, used Si force constants to approximate those for Ge and interfacial interactions[18]. We note that all these previous studies modeled sharp interfaces without

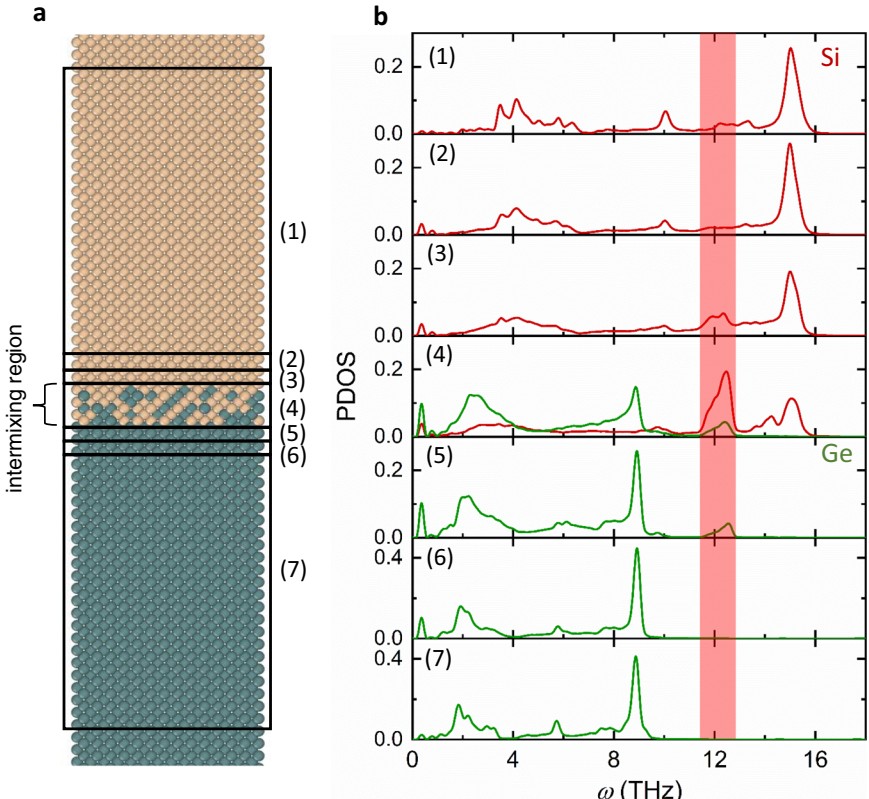

**Fig. 3 Molecular dynamics (MD) simulations confirming interfacial modes. a** Schematic diagram of the MD-simulated system and the regions where phonon density of states (PDOS) are calculated. **b** PDOS by MD simulations at different locations throughout the domain along the direction perpendicular to the Si-Ge interface. Each panel is the PDOS of the atoms at a location at or away from the interface (middle of the interfacial mixture) indicated in **a**. Interfacial modes exist at the intermixing region and the first adjacent atomic layer on each side.

considering atomic mixing effects. For comparison, we also performed simulations on an ideally sharp Si-Ge interface and also found a peak around 11.6–12.7 THz (Supplementary Fig. 7).

To understand how these interfacial modes contribute to interfacial thermal transport, we performed non-equilibrium MD (NEMD) simulations to calculate the TBC (see "Methods" section and SI for details). The calculated TBC is ~250 MW m$^{-2}$ K$^{-1}$ at 300 K. We further used spectral analysis to calculate the contributions of phonons with different frequencies to the TBC (see "Methods" section and SI for details). This does not only allow us to directly visualize the role of interfacial modes, but also allows quantum correction to be applied so that the MD-calculated TBC can be directly compared to our experimentally measured TBC. As shown in Fig. 4a, there is an obvious peak of the frequency-dependent TBC around the interfacial phonon mode region (11.4–12.8 THz) from our spectral analysis. As MD are classical simulations, all phonons are equally excited. However, in reality, phonon follows the Bose–Einstein distribution. Such an error can be corrected by applying a quantum correction by weighing the contributions of different frequencies with the ratio of the quantum and classical heat capacities of that frequency (see "Methods" section and SI for details). The quantum correction leads to reduced contributions from high-frequency modes as they are not fully excited, but the contributions from the interfacial modes are still obvious. We integrate the frequency-dependent TBC to obtain the cumulative TBC, and find that the interfacial modes contribute ~5% of the total TBC at 300 K (Fig. 4b). The quantum-corrected TBC (231 MW m$^{-2}$ K$^{-1}$) agrees very well with our TDTR measurements (244 and 236 MW m$^{-2}$ K$^{-1}$ by different TDTR systems). We note that these calculations are on the mixed interface. For a sharp

interface, the calculated TBC is 197 MW m$^{-2}$ K$^{-1}$, which is lower than the mixed interface and farther away from the measured TBC. This is consistent with previous findings that interfacial mixing can enhance TBC[11,12]. Figure 4c also show the comparison of TBC values of Si-Ge interfaces calculated from DMM, AGF (both harmonic and anharmonic), and previous MD simulations using different potentials. The NNP-MD results for both the mixed interface and the sharp interface respectively agree well with their counterparts from AGF calculations, highlighting the importance of correctly capturing the microscopic interface conditions for TBC predictions. On the other hand, other models (e.g., DMM) that ignore such microscopic interfacial details show inferior agreement with experimentally measured TBC. The calculation details of non-equilibrium Landauer approach based on DMM can be found in the "Methods" section and SI. The fact that NNP-MD is accurate in predicting TBC also gives confidence on its predicted non-trivial contributions of interfacial modes to the TBC.

We further performed temperature-dependent TDTR measurements and MD simulations from 300 to 500 K. The TDTR setup we used for high-temperature measurements is at University of Illinois at Urbana-Champaign (UIUC). The measured TBC matches the value measured by the TDTR at Georgia Institute of Technology (GT) at 300 K, as shown in Fig. 5a. The measured TBC values also agree well with the MD-calculated TBC values with NNP, which again confirms the accuracy of the present NNP for the Si-Ge interface. Both the measured and calculated TBC values show a weak increase with increasing temperature, indicating that the population of the phonons involved in interfacial thermal transport only increases slightly in this temperature range —a trend commonly seen in the

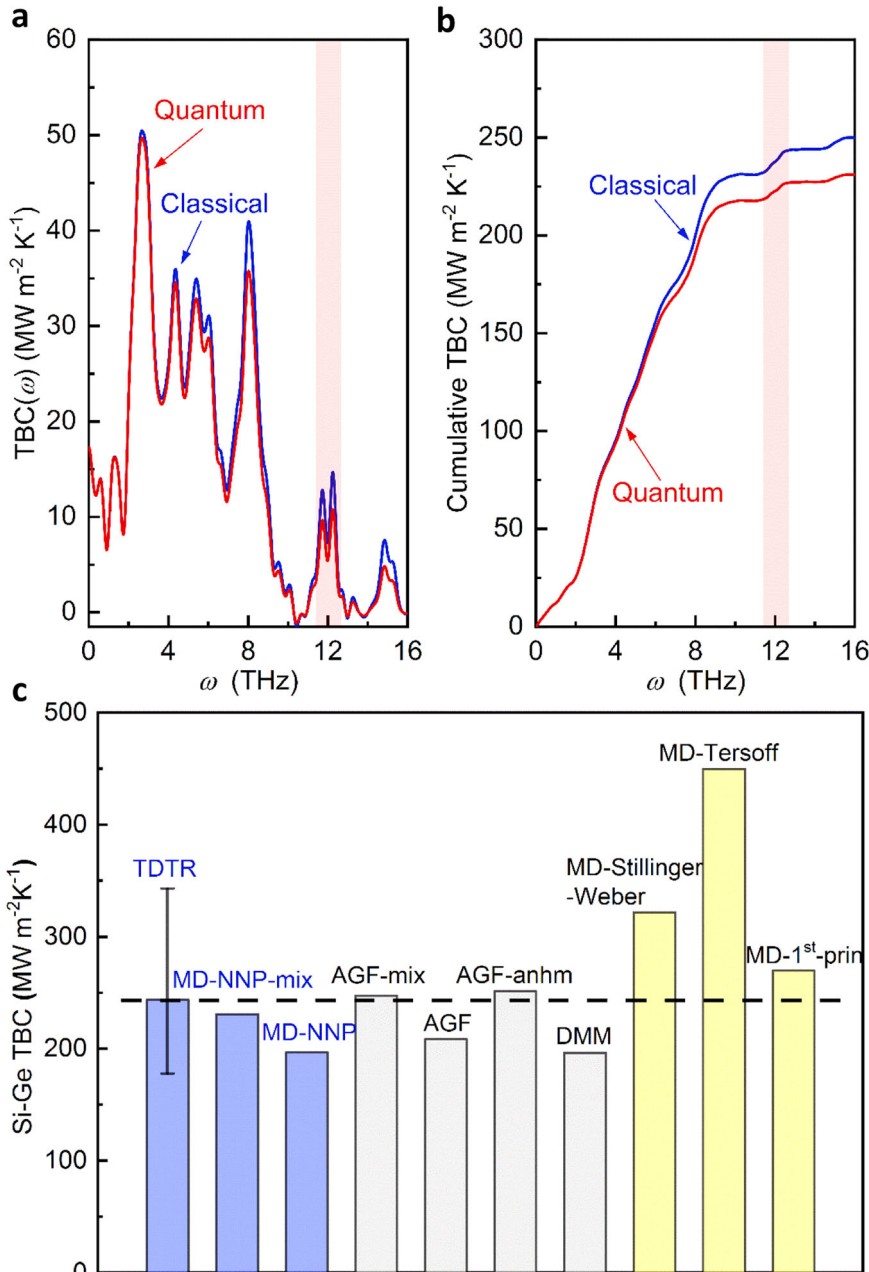

**Fig. 4 Thermal boundary conductance (TBC) of Si-Ge interfaces at 300 K. a** Spectral TBC of the Si-Ge interface. **b** Cumulative TBC of the Si-Ge interface. **c** Comparison of the measured TBC with calculated TBC values. The error bar for the measured TBC is calculated by a Monte Carlo method, which considers all the possible error sources (see SI). The "MD-NNP-mix" and "MD-NNP" are TBC values of mixed and ideal Si-Ge interfaces by MD with NNP. "AGF" and "AGF_mix" are TBC values of an ideal Si-Ge interface and a Si-Ge interface with 6-atomic-layer mixing calculated by AGF[11]. "AGF_anhm" is the TBC of a perfect Si-Ge interface calculated by AGF, which includes both harmonic and anharmonic contributions[13]. "DMM" is the TBC calculated by the non-equilibrium Landauer approach, which considers the non-equilibrium effect at the interface and uses DMM to calculate transmission. The TBC of perfect Si-Ge interfaces calculated with other interatomic potentials such as Stillinger–Weber potential, Tersoff potential, and first-principles-calculated force constant potential are also included for comparison[18,19,27,28].

temperature-dependent TBC of many other solid interfaces at high temperatures.

To compare with the interfacial mode contributions to TBC at 300 K, spectral analysis is also conducted for the interface at 460 K with MD and the quantum-corrected results are shown in Fig. 5b, c. The two accumulation curves at different temperatures in Fig. 5c share a similar trend, with an obvious jump around the frequency of the interfacial mode (11.4–12.8 THz). Although the absolute contributions from the interfacial modes increase with

increasing temperature, the relative contributions remain around 5% as the overall TBC also increases slightly at 460 K.

It is noted that the frequencies of the interfacial modes are higher than those of the optical phonons of bulk Ge. Elastic processes across interfaces require that bulk phonon modes on both sides have the same frequency. Therefore, for the Si-Ge interface, phonons with frequencies above ~10 THz cannot contribute to TBC via elastic processes while they can contribute to TBC via inelastic processes. Inelastic processes allow phonons

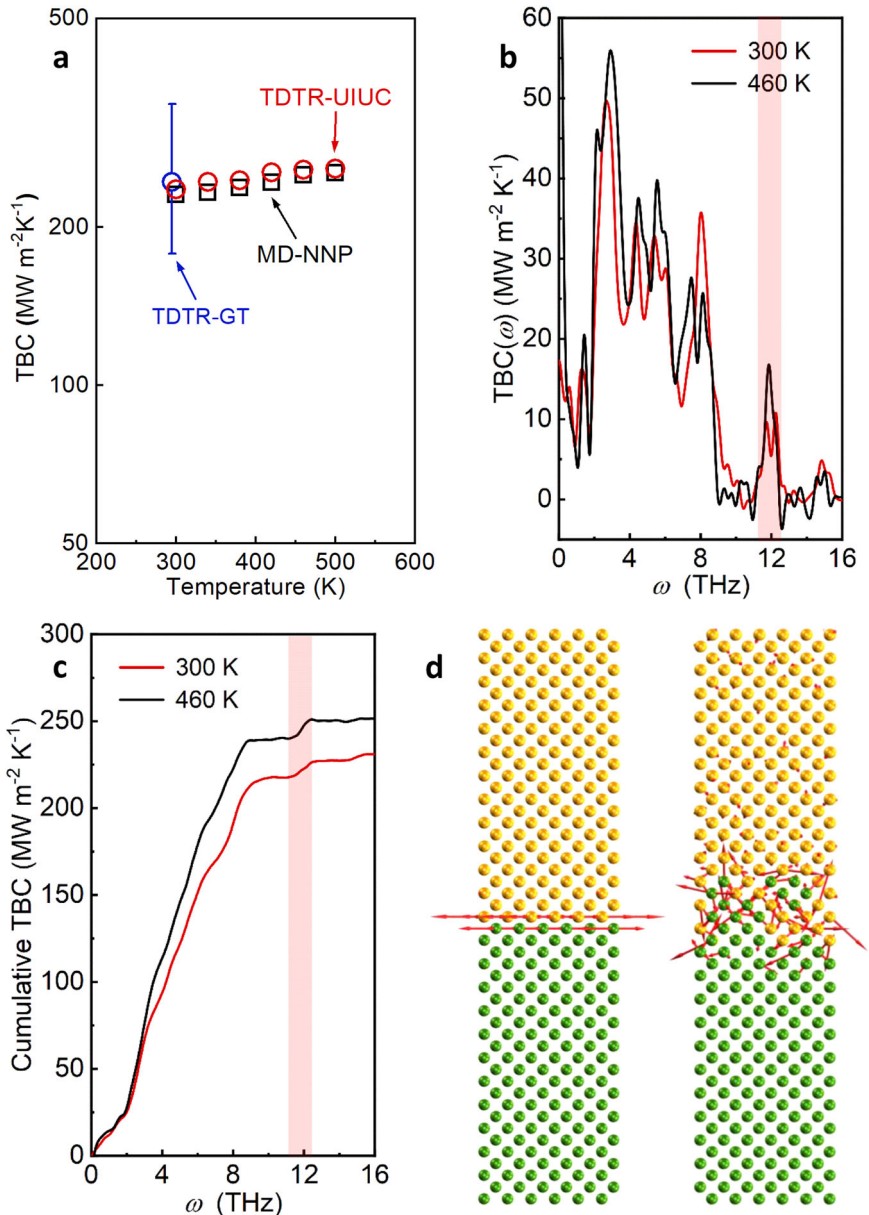

**Fig. 5 Temperature-dependent interfacial thermal transport. a** Temperature-dependent thermal boundary conductance (TBC) of Si-Ge interfaces measured by the TDTR systems at University of Illinois Urbana-Champaign (UIUC) and Georgia Institute of Technology (GT), and calculated by MD simulations. The error bar of the measured TBC is calculated by a Monte Carlo method, which considers all the possible error sources (see SI). **b** Comparison of MD-calculated spectral TBC of the Si-Ge interface at 300 K and 460 K. **c** Comparison of accumulative TBC of the Si-Ge interface at 300 K and 460 K. **d** Eigenvectors for an interfacial mode at ~12.4 THz for a sharp interface (left) and the interface with interfacial mixing (right). The upper side is Si and the lower side is Ge.

with different frequencies to exchange energy and contribute to TBC[18,28–31]. Previous simulations have shown that the role of the interfacial modes localized at the interface is to act as a bridge coupling the bulk phonon modes on both sides, which facilitates the inelastic transport across the interfaces for phonons with different frequencies[17,28,32]. We note that although we still call this an inelastic process, because energy is transferred from one frequency to another across the interface, the underlying mechanism is not originated from a conventional phonon scattering picture. Instead, it is due to the correlation between the localized interfacial phonons and other phonons that extend into the materials making up the interface[17,32]. It is notable that the MD simulations with NNP in this work intrinsically include both elastic and inelastic processes. However, other theoretical models

such as DMM, AMM, and AGF typically only consider the elastic phonon transport even though the inelastic processes can contribute to the total TBC[3,28,33]. The recent development of AGF enables the consideration of both elastic and inelastic contributions but it is very computationally expensive[13].

To explore the physical origin of the interfacial modes, we visualize the eigenmodes of an example interfacial mode at ~12.4 THz near the interface with and without atomic mixing (Fig. 5d). It can be seen that the eigenvectors are mainly localized around interfacial atoms that involve Si-Ge bonds for both the sharp and mixed interfaces. For the mixed interface, the delocalization length of the interfacial modes is around 1.1 nm in the interfacial region, which agrees well with that of the EELS-detected interfacial modes (~1.2 nm around the interface) in

Fig. 2d. The origin of the interfacial modes is thus believed to be from the Si-Ge bonds that do not exist in bulk Si or bulk Ge. It has been reported that the Si atoms encapsulated in Ge atomic cages can lead to modes at a frequency that does not exist in bulk Si and Ge, and the observed frequency is similar to that of the interfacial modes[34]. This suggests that the special modes from the Si-in-Ge cage structure and interfacial modes have the same origin, which is the Si-Ge bonds.

In summary, we experimentally observed the existence of localized interfacial vibrational modes within 1.2 nm at an epitaxial Si-Ge heterointerface by Raman spectra and high-energy-resolution EELS in a STEM. These interfacial modes are found to be reproducible using MD simulations with a high-fidelity NNP, which also yield TBC agreeing favorably to TDTR measurements. The spectral analysis in MD indicates that these interfacial modes, which are mainly localized near the interface, can have non-trivial contributions to TBC. This work paves the way to fundamentally understanding heat transport across realistic interfaces and may stimulate the development of theories of interfacial thermal transport. It will impact applications such as electronics thermal management and thermoelectric energy conversion.

## Methods

### Materials growth
The two samples in this work were grown by MBE. The silicon substrates (boron doped $5 \times 10^{14}$–$1 \times 10^{15}$ atoms/cm$^3$) were given a modified Radio Corporation of America cleaning procedure followed by an HF dip as the last step. Then, the samples were immediately loaded into the vacuum systems and pumped down from atmospheric pressure to $<10^{-6}$ Torr in 15 min. Details can be found in the literature[35]. For Sample 1 (10 nm Si-300 nm Ge-Si substrate), the temperature of the silicon substrate (boron doped $5 \times 10^{14}$–$1 \times 10^{15}$ atoms/cm$^3$) was first increased to 700 °C in 3 min. A 25 nm Si buffer was grown on the Si substrate at 700 °C, followed by a 300 nm Ge layer in three parts (10 nm Ge at 300 °C, 80 nm Ge at 400 °C, and annealed at 700 °C, then a 210 nm Ge at 400 °C), and finally a 10 nm Si cap was grown on the Ge at 400 °C. The annealing time of the Ge layer at 700 °C is 16.67 min (1000 s). The cooling time after the Ge annealing is ~6 min (~1 °C/s from 700 °C to 400 °C). Similarly, the cooling rate from the Si homo-epitaxial layer to the initial Ge growth layer is ~ 1 °C/s with a 2 min settling time (~8 min in total). Sample 2 (250 nm Ge-Si substrate) was grown in a similar way but without the 700 °C annealing process. The growth temperature of the 250 nm Ge layer was dropped over an 8 min period to ~400 °C, where the 250 nm Ge was grown at a constant temperature. More details can be found in SI.

### Raman spectroscopy
Raman measurements were performed on Sample 1, a Si wafer, and a Ge wafer with a Renishaw InVia Raman system. The Ar$^+$ laser wavelength is 488 nm. The objectives are Leica 0.50 NA ×50 and 0.75 NA ×100 with theoretical diffraction-limited spot sizes of 1.19 μm and 0.79 μm, respectively. The ×50 and ×100 objectives were used to measure Sample 1. The ×50 objective was used to measure the reference Si and Ge wafers. The acquisition time of each Raman spectrum is 30 s with five accumulations.

### Thermal characterizations
The TBC of Sample 2 were measured by TDTR. TDTR is an ultrafast-laser-based pump-probe technique which is capable of measuring thermal transport properties of both nanostructured and bulk materials[36,37]. A modulated pump beam heats the sample surface periodically while a delayed probe beam detects the temperature variation of the sample surface. The signal picked up by a photodetector and a lock-in amplifier was fitted with an analytical heat transfer solution of the sample structure to infer unknown thermal properties. An 80 nm Al layer was deposited on the sample as the TDTR transducer. For the TDTR measurements at GT, a ×10 objective was used for all the measurements with a pump radius of 10.7 μm and a probe radius of 5.8 μm. The modulation frequency is 8.8 MHz. For the TDTR measurements at UIUC, a ×5 objective was used for all the measurements with a pump/probe radius of 10.7 μm. The modulation frequency is 9.3 MHz. More details about thermal measurements can be found in the SI.

### TEM measurements
HAADF-STEM was performed using JEOL JEM-ARM300CF S/STEM, equipped with double aberration correctors with a spatial resolution of ~0.6 Å at 300 kV. HAADF-STEM images were recorded using a convergence semi-angle of 22 mrad and collection semi-angles at 83–165 mrad.

### EELS measurements
The vibrational spectra were performed using a Nion UltraSTEM 200 microscope operating at 60 kV. The convergence semi-angle was 33 mrad and the probe current was about 120 pA, offering a spatial resolution of

1.5 Å. The probe current needs to remain high to warrant adequate vibrational signals on the EELS detector. The EELS collection semi-angle was 25 mrad to allow all phonon branches inside the entire BZ to be recorded by EELS. With the implementation of an alpha-type monochromator and designed magnetic-prism spectrometer, the energy resolution of EELS is routinely 7–8 meV (1.7–1.9 THz) with an exposure time of 1 s[38]. High spatial-resolution EELS line-scan datasets were automatically performed and assembled by running our custom-developed python script on top of Nion's Swift software. To optimize the signal-to-noise ratio and mitigate the influence of high voltage instability, the 40-pixel line-scan data in Fig. 2b with a step size of 0.15 nm was acquired by summing 50 frames with 1 s exposure per frame, which were then all aligned by the center of zero-loss peak. Besides, the single point spectra in Fig. 2c were obtained by summing 200 frames with 1 s exposure per frame, which were captured separately from the line-scan. The EELS dispersion was about 0.3 meV/channel. Background subtraction was carried out for each spectrum by fitting an exponential polynomial function.

### Non-equilibrium Landauer approach
Phonons near the interfaces are in strong non-equilibrium conditions because of the difference in modal transmission coefficients and reservoir temperatures. By considering this non-equilibrium effect, a corrected non-equilibrium Landauer approach was developed to calculate TBC[3,39]. Here, DMM was used to calculate the transmission. A detailed comparison of non-equilibrium Landauer approach and conventional Landauer approach is included in the SI.

### NNP construction
To build a NNP for the Si-Ge interfaces, we first generated a training dataset consisting of snapshots from ab initio MD (AIMD) simulations[40]. Each snapshot contains the data of potential energy, atomic forces, atomic coordinates, and supercell lattice vectors. An $8 \times 3 \times 3$ Si-Ge interface supercell, with a Si lattice constant ($a = 5.4307$ Å), is simulated in the canonical ensemble (NVT) at temperatures ranging from 50 to 600 K. The system is equilibrated for 1.0 ps with a time step of 1.0 fs and the snapshots are then collected every five steps during 4.0 ps-long production runs. To capture the effect of interfacial mixing, we also generated a dataset for two different $4 \times 3 \times 3$ Si-Ge crystalline alloys with the Si lattice constant adopting the same procedure. AIMD simulations are carried out using the QUICKSTEP algorithm implemented in the CP2K package, which leverages the Gaussian and plane waves approaches[41,42]. We employ the Goedecker–Teter–Hutter (GTH) double-ζ, single polarization (DZVP-MOLOPT-GTH) basis set, and the GTH-PBE pseudopotentials[43] to describe the core-valence interactions. The wave-plane cutoff is set to be 300 Ry and the BZ is sampled at the Γ point. Finally, the training dataset contains 9600 snapshots for a perfect Si-Ge interface and 8000 snapshots for the Si-Ge crystalline alloy mixtures.

The NNP is obtained by training a deep neural network potential scheme developed by Zhang et al.[44,45], which has been proved to be capable of simulating Pt/MoS$_2$ interfaces[46]. This scheme comprises an embedding network and a fitting network. The embedding network first maps the chemical species and atomic coordinates to the embedded features that preserve all the natural symmetries. Then a three-layer feed-forward network learns the potential energies and forces by minimizing the mean squared error loss during training on the ab initio dataset. The interaction cutoff for this potential is set to be 6 Å. More details can be found in the SI.

### NEMD simulations
NEMD simulations were performed using the LAMMPS package[47] with this NNP for the Si-Ge interfaces. For both Si and Ge sides, the number of conventional cells along the x, y, and z directions are chosen to be equal to 25, 7, and 7, respectively (see SI for details). A 0.7 nm-long mixed region is formed by randomly shuffling the atoms near the interface to mimic our measured Si-Ge interfaces. The interface is a plane perpendicular to the x-direction, which is at the middle of the interfacial mixture and perpendicular to the [100] crystallographic direction. Periodic boundary conditions are applied to all three spatial directions, and a time step of 1.0 fs is used for the MD simulations. After relaxing the structure in the NVT ensemble for 0.5 ns, we freeze the two ends of the system and apply a temperature difference using Langevin thermostats in the microcanonical (NVE) ensemble for 2.0 ns. The temperatures on the two thermostats are ±25 K of the target mean temperature. The TBC is obtained by $G_{\text{Si}-\text{Ge}} = Q/(A \triangle T)$, where $Q$ is the steady-state heat flux along the x-direction, $A$ is the cross-sectional area, and $\Delta T$ is the temperature difference at the interface determined by extrapolating linear fits of the temperatures of the two sides to the interface and calculating the difference. Considering that classical statistics overestimates modal heat capacity at high frequencies[48], it is necessary to apply quantum correction to these predictions. To account for the quantum effect, we need to quantify the contributions to the interfacial thermal transport on the mode level. The spectral analysis[49–51] is employed to determine the modal contributions to TBC. With the modal decomposition of TBC, we apply quantum correction by multiplying $G(\omega)$ by the factor $u^2 e^u/(e^u - 1)^2$, where $u = \hbar\omega/k_B T$ represents the ratio of the quantum heat capacity to the classical one. More details can be found in the SI.

**Reporting summary**. Further information on research design is available in the Nature Research Reporting Summary linked to this article.

## Data availability
The datasets generated during and/or analyzed during the current study are available from the corresponding authors upon reasonable request.

## Code availability
The code used for calculations, simulations, and data analysis is available from the corresponding authors upon reasonable request.

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

## Acknowledgements

We acknowledge the financial support from Office of Naval Research MURI Grant Number N00014-18-1-2429. We thank helpful discussions with Asegun Henry, Andrew

Rohskopf, and Tianli Feng. The TEM work was supported by the Department of Energy (DOE), Office of Basic Energy Sciences, Division of Materials Sciences and Engineering under Grant DE-SC0014430, and partially supported by the Center for Nanophase Materials Sciences, which is a DOE Office of Science User Facility (JCI). We acknowledge the use of facilities and instrumentation at the UC Irvine Materials Research Institute (IMRI), which is supported in part by the National Science Foundation through the UC Irvine Materials Research Science and Engineering Center (DMR-2011967).

## Author contributions

Z.C., S.G. and T.L. initialized the project. Z.C. and N.J.H. did the Raman measurements. R.L., E.L. and T.L. finished the NNP development and NEMD simulations. X.Y., C.G., X. P. and J.C.I. performed the TEM and EELS measurements. Z.C. performed the TDTR measurements. G.J. and K.D.H. grew the samples. J.S. did the non-equilibrium Landauer calculations. M.E.L. and M.S.G. performed the XRD measurements. Z.C. coordinated the project and wrote the manuscript with inputs from all authors. X.Y. and R.L. assisted with manuscript preparation, prepared figures, and reviewed the manuscript. T.L. provided insights in understanding the data and edited the manuscript. S.G. initially motivated the investigation, provided overall guidance to the project, and reviewed the manuscript.

## Competing interests

The authors declare no competing interest.
