## [Peer Review File · Nature Communications]

Experimental Observation of Localized Interfacial Phonon ModesREVIEWER COMMENTS

Reviewer #1 (Remarks to the Author):

Review report for the manuscript titled, "Experimental Observation of Localized Interfacial Phonon Modes" by Z. Cheng et al.

The authors report the experimental observation of localized interfacial phonon modes at Si-Ge interfaces formed by molecular beam epitaxy. The authors perform Raman and electron energy-loss spectroscopy (EELS) measurements on one of the samples to observe the signatures of these localized modes. They also perform time-domain thermoreflectance measurements of the thermal boundary conductance (TBC) at the Si-Ge interface of another sample, and use non-equilibrium molecular dynamics (NEMD) simulations with neural network potentials (NNP) to shed light on the measured TBC.

This work is an exciting experimental demonstration of previous theoretical predictions by others in the literature. While I cannot recommend acceptance of the current version of the manuscript, I recommend consideration of a revised version (after appropriately addressing my comments below) for possible publication in Nat. Comm.

Comments:

1. The TDTR measurement is performed only at 300 K. Although the measurements seem to be in reasonable agreement with the NNP-NEMD calculation, I feel that a lot more insight can be obtained from temperature-dependent measurements (and calculations) of TBC. In particular, I am interested to know if these interfacial modes get occupied in the same way as a bulk acoustic/optic mode with temperature. Also, what are their energy transport velocities?
2. The observed contribution from these interfacial modes to the overall TBC seems to be tiny ($\sim 5\%$). Perhaps, at higher temperatures, the contributions could be greater? Another incentive to perform measurements at other temperatures!
3. The sensitivity to the TBC is lower than that to the thermal conductivity of Ge for the most part in the supplementary figure 2. Perhaps this issue can also be resolved by performing measurements at other temperatures?
4. The sample used for Raman and EELS measurements are different from that used for TDTR measurements. Is it possible at all to perform all measurements on a single sample?
5. In fig. 4(a), there are contributions to TBC in the frequency range of 14-16 THz (outside the region marked as interfacial modes). Naively, I would think that, since Ge has a max. optic phonon frequency of ~ 10 THz, there shouldn't be any bulk-like modes contributing to TBC. Can the authors comment on the contribution to TBC from this frequency range?
6. In fig. 2(b), the dotted vertical lines and the dashed rectangle make it difficult to clearly see the (already weak) intensity around 11-12.5 THz in the interfacial region. Moreover, I am able to see similar intensities in the same frequency region away from the interface on the Si side (particularly, between "-2" and "-3" on the "Distance to interface (nm) axis). Can the authors comment on the origin of these intensities as well?

Reviewer #2 (Remarks to the Author):

Manuscript NCOMMS-21-25101-T

"Experimental Observation of Localized Interfacial Phonon Modes"
by Zhe Cheng et al.

Referee Report

The paper is devoted to the experimental Raman and EELS-STEM studies of the vibrational (phonon) modes confined at the interface between Ge and Si lattices. The paper reports the first experimental observation of localized interfacial phonon modes at ~ 12 THz at a high-quality epitaxial Si-Ge interface. These modes are further confirmed by the molecular dynamics simulations of the thermal boundary conductance through such interface, which is measured in time-domain thermoreflectance experiments.

The paper presents interesting and comprehensive results, confirming by different means the existence of the phonon modes confined at the Si-Ge interface. But the physical origin of such modes is not revealed and discussed, which lowers the merit of the paper. The observed interface modes, which are out of the phonon band in Ge crystal, can be related with the vibrations of the light Si atoms that are encapsulated in the effective cages by the surrounding heavy Ge atoms. Such atomic configurations are present in the interfacial mixing domain shown in Fig. 3 and Supplementary Figs. 6 and 7. On the other hand, the resonance phonon mode and transmission at ~ 13 THz (within the Tersoff interatomic potential) was revealed, e.g., in AIP Conference Proceedings 2241, 020023 (2020), doi: 10.1063/5.0011386, in which phonon transmission through the monolayers of Si atoms, symmetrically encapsulated by the layers of Ge atoms in Si lattice, was studied and different effective cages for light Si atoms were described.

In my opinion, the paper can be published in Nature Communications with the proper discussion of the physical origin of the observed interface phonon modes.

Reviewer #3 (Remarks to the Author):

Comments on NCOMMS-21-25101-T

Title: "Experimental Observation of Localized Interfacial Phonon Modes"

Authors: Zhe Cheng, *et al.*

The authors, for the first time, confirm the existence of interfacial phonon modes by creating high-quality Si-Ge interfaces, using molecular beam epitaxy and detecting the localized modes by Raman spectroscopy and high-energy-resolution electron energy-loss spectroscopy in a scanning transmission electron microscope. These modes are further confirmed using molecular dynamics simulations with a high-fidelity neural network interatomic potential, which also yield TBC agreeing well with that measured from time-domain thermoreflectance (TDTR) experiments. It is an interesting work, however, the present is in poorly written and lacks in-depth analysis. Besides, there are several logical problems existing in the work, and some of anomalous experimental data need to be confirmed. Therefore, a significant modification should be done for current manuscript for further consideration of publication in *Nature Communications*.

1. As mentioned in the "Introduction" section, "For insulator or semiconductor-related interfaces, lattice vibrations (phonons) are the dominant heat carriers for interfacial thermal transport ", further acoustic phonons rather than optical typically play the predominated role in heat transfer based on the Debye-Callaway model. Nevertheless, peaks indexed to interfacial modes are located at ~11.4-12.8 THz for both Si and Ge atoms shown in Fig. 3, and "The PDOS peaks of Si and Ge optical phonons are ~15 THz and ~9 THz", thus these additional phonons should belong to optical frequency branch, and would not affect the thermal properties evidently. Therefore, please explain why the interfacial modes "can have non-trivial contribution to TBC"?
2. On page 9, "Clear peaks of the vibrational spectral intensity are observed around the interface at 11.6 and 12.0 THz", however, the peaks are not clear in Fig. 2d, especially for that of 12.0 THz. Therefore, the evidence based on EELS-STEM is not sufficient, please provide more analysis here and take a deep insight. Otherwise, the existence of interfacial modes should be questioned to some extent.
3. Is there any experimental evidence of TEM or STEM for the validity of atomic mixing effects shown in Fig. 3(a)? After all, no such difference can be

found in the HAADF-STEM image, and sharp interfaces are typically used by previous studies.

4. How to calculate the TBC? The details shown in “Methods” section are confusing and should be clear for this calculation process.

5. Additional contents should be added.

1) On page 6, “Sample 2 is a 250-nm Ge layer grown on a Si substrate and is used for TBC measurements, where the Ge layer thickness is optimized to maximize the sensitivity of the TBC of the Si-Ge interface in TDTR measurement.” The optimization process of Ge layer thickness and TDTR measurements require intuitive data.

2) On page 18, “The two samples in this work were grown by molecular beam epitaxy (MBE). The silicon substrates (boron doped 5×10^{14} - 1×10^{15} atoms/cm³) were given a modified RCA cleaning procedure followed by an HF dip as the last step”, please provide the full name of “RCA”.

6. There are many grammar errors existing in this work, the paper should be thoroughly polished.

1) On page 3, “Simulations find that the interfacial phonon modes have obvious contribution to the total TBC”, “an” should be added before “obvious”.

2) On page 4, “Interfaces impede heat flow in micro/nanostructured systems, which highlight the growing importance of thermal management of microelectronics and devices for energy conversion and storage”, “highlight” should be corrected as “highlights”.

3) On page 16, “highlighting the important of correctly capturing the microscopic interface conditions”, “important” should be modified as “importance”; “The fact that NNP-MD is accurate in predicting TBC also give confidence on its predicted non-trivial contribution”, “give” should be replaced with “gives”.

4) Lots of confusing sentences should be optimized, *e.g.*, on page 5, “several molecular dynamics (MD) studies have shown that phonon modes that are unique to the interface (i.e., interfacial phonon modes) can play important roles in interfacial thermal transport”; “these MD simulations were either based on empirical interatomic potentials that were not specifically developed for interfaces or employed approximation for interfacial interactions despite the use of first-principles force constants”.

...

We sincerely thank the reviewers and the editor for the precious time and attention on our manuscript. We also greatly appreciate the constructive and valuable comments from the reviewers. These insightful suggestions and corresponding revisions improve our manuscript and make it more complete.

Reviewer #1:

The authors report the experimental observation of localized interfacial phonon modes at Si-Ge interfaces formed by molecular beam epitaxy. The authors perform Raman and electron energy-loss spectroscopy (EELS) measurements on one of the samples to observe the signatures of these localized modes. They also perform time-domain thermoreflectance measurements of the thermal boundary conductance (TBC) at the Si-Ge interface of another sample, and use non-equilibrium molecular dynamics (NEMD) simulations with neural network potentials (NNP) to shed light on the measured TBC.

This work is an exciting experimental demonstration of previous theoretical predictions by others in the literature. While I cannot recommend acceptance of the current version of the manuscript, I recommend consideration of a revised version (after appropriately addressing my comments below) for possible publication in Nat. Comm.

Response: We thank the reviewer for the positive evaluation of our manuscript as “exciting experimental demonstration”.

Comment (1): The TDTR measurement is performed only at 300 K. Although the measurements seem to be in reasonably agreement with the NNP-NEMD calculation, I feel that a lot more insight can be obtained from temperature-dependent measurements (and calculations) of TBC. In particular, I am interested to know if these interfacial modes get occupied in the same way as a bulk acoustic/optic mode with temperature. Also, what are their energy transport velocities?

Response: We thank the reviewer for the insightful comment. In the revision, we have performed temperature-dependent TDTR measurements and MD simulations at 300-500 K. The TDTR setup

we used for high temperature measurements is at University of Illinois at Urbana-Champaign (UIUC). The measured TBC at room temperature matches well with the value measured by the TDTR at Georgia Institute of Technology (GT).

Fig. 5 Temperature dependent interfacial thermal transport. **a** Temperature dependent TBC of Si-Ge interfaces measured by the TDTR systems at University of Illinois Urbana-Champaign

(UIUC) and Georgia Institute of Technology (GT), and calculated by MD simulations. **b** Comparison of MD-calculated spectral TBC of the Si-Ge interface at 300 K and 460 K. **c** Comparison of accumulative TBC of the Si-Ge interface at 300 K and 460 K. **d** Eigenvectors for an example interfacial mode at ~ 12.4 THz for a sharp interface (left) and an interface with interfacial mixing (right). The upper side is Si and the lower side is Ge.

As shown in Fig. 5a above, the measured TBC values of the Si-Ge interface with different TDTR systems agree well. The measured TBC also matches well with the MD-calculated TBC with NNP, which also confirms the accuracy of the present NNP for the Si-Ge interface. Both the measured and calculated TBC values show a weak increase with increasing temperature, which indicates that the population of the phonons involved in interfacial thermal transport only increases slightly in this temperature range – a trend commonly seen in the temperature dependent TBC of many other solid interfaces at high temperatures.

The interfacial modes get occupied in a similar way as optical modes.¹ At a given temperature, the phonon distribution follows Bose-Einstein distribution which only depends on the phonon frequency. We note that the interfacial modes, which are mainly localized around the interface and have a high frequency of ~ 12 THz, are originated from the Si-Ge bonds unique to the interfacial region. The vibration of this bond is an intra-unit cell phenomenon, so it is similar to the origin of optical phonons. We also calculate the vibrational modes by finite difference method, and Fig. 5d shows the eigenvectors of an example interfacial mode at ~ 12.4 THz for a sharp interface and an interface with atomic mixing. It is found that the eigenvectors are mostly localized in the mixed region, and only a few of them spatially extend to the first or second layer next to the intermixing layers, which is consistent with the PDOS shown in Fig 3. Therefore, energy transport velocity is not well-defined for these interfacial modes, which are mainly localized near the interface, and the energy is transferred across the interface mainly by the coupling between interfacial modes and other propagating modes.^{2,3}

Revisions: We added Fig. 5 to the main text. The related discussions were added in the main text (Page 18) and Supplementary Information (Page 12).

Comment (2): The observed contribution from these interfacial modes to the overall TBC seems to be tiny (~5 %). Perhaps, at higher temperatures, the contributions could be greater? Another incentive to perform measurements at other temperatures!

Response: We thank the reviewer for the suggestion. We agree with the reviewer and performed TDTR measurements and MD calculations at high temperatures, as discussed in the response of Comment (1). To compare with the data at 300 K, spectral analysis is also conducted for the Si-Ge interface at 460 K with NNP. Fig. 5b shows the contributions from different phonon modes after quantum correction at 300 K and 460 K, and Fig. 5c shows the cumulative TBC as a function of frequency. The two accumulation curves at different temperatures in Fig. 5c share a similar trend, with an obvious jump around the frequency of the interfacial mode (11.4-12.8 THz). While the absolute contributions from the interfacial modes increase with increasing temperature, the relative contributions remain around 5% as the overall TBC also increases slightly at 460 K.

Revisions: We added the related discussions to Pages 18-19 in the main text.

Comment (3): The sensitivity to the TBC is lower than that to the thermal conductivity of Ge for the most part in the supplementary figure 2. Perhaps this issue can also be resolved by performing measurements at other temperatures?

Response: We thank the reviewer for the helpful suggestion. The figure below shows the sensitivities of Si-Ge TBC and Ge thermal conductivity as a function of delay time and temperature. The sensitivity of Si-Ge TBC decreases as temperature increases while the sensitivity of Ge thermal conductivity increases slightly as temperature increases. Typically, the TDTR measurements are reliable once the sensitivity of the unknown parameter is larger than 0.1. The TBC and Ge thermal conductivity values measured by the two TDTR systems are consistent. Please note that the error bar of the measured Si-Ge TBC calculated by the Monte Carlo method in this work considers all the possible error sources, including the effect from the Ge thermal conductivity in the data fitting. The error bars are similar for both TDTR systems at room temperature. For high temperature measurements, the sensitivity of TBC decreases. But the errors from other parameters become smaller, such as the heat capacity of each layer and the Si thermal

conductivity, because heat capacity values are less sensitive to temperature at high temperatures and Si thermal conductivity is less sensitive to defect scatterings due to increased anharmonic scattering. We thus obtained similar error bars for the reported TBC values at high temperatures. As shown in Supplementary Fig. 8c, the thermal conductivity of bulk Ge is also included for comparison.⁴ The thermal conductivities of both the Ge thin film and bulk Ge decrease as temperature increases due to increased phonon-phonon scatterings. The reduced thermal conductivity of the Ge thin film compared to thermal conductivity of bulk Ge is due to size effect.

Supplementary Fig. 8 Temperature dependent thermal measurements. **a** Sensitivity of Ge thermal conductivity as a function of delay time and temperature. **b** Sensitivity of Si-Ge TBC as a function of delay time and temperature. **c** The measured thermal conductivity of the 250-nm Ge thin film (TDTR-GT and TDTR-UIUC) and bulk Ge at different temperatures. The reduction in thermal conductivity of the Ge thin film compared to the bulk thermal conductivity is due to size effect.⁴

Revisions: We added Supplementary Fig. 8 and related discussions to Pages 13-14 in the Supplementary Information.

Comment (4): The sample used for Raman and EELS measurements are different from that used for TDTR measurements. Is it possible at all to perform all measurements on a single sample?

Response: We thank the reviewer for this comment. The Raman measurements need optical access to the interface so the top Si layer needs to be very thin. On the other hand, TDTR measurements need a thick Ge layer. These factors prevent us from performing all the measurements on a single sample. However, the epitaxial Si-Ge interfaces are expected to be similar. To show that the two interfaces are similar, we characterized the Si-Ge interface of the TDTR sample using atomic-resolution STEM imaging, as shown in the figure below. The Si-Ge interface of the TDTR sample has a similar epitaxial growth structure as the interface used for Raman and EELS measurements.

Supplementary Fig. 9 Interfacial structures of the two samples. a Atomic resolution STEM image of the Si-Ge interface for the Raman and EELS measurements. **b** The annular dark-field (ADF) intensity to show the atomic mixing at the interface. **c** Atomic resolution STEM image of

the Ge-Si interface of the sample for TDTR measurements. **d** The ADF intensity across the interface.

Revisions: We added Supplementary Fig. 9 and related discussions to Pages 15-16 in the Supplementary Information.

Comment (5): In fig. 4(a), there are contributions to TBC in the frequency range of 14-16 THz (outside the region marked as interfacial modes). Naively, I would think that, since Ge has a max. optic phonon frequency of ~10 THz, there shouldn't be any bulk-like modes contributing to TBC. Can the authors comment on the contribution to TBC from this frequency range?

Response: This is a very good question. Those are the inelastic contributions to TBC. Elastic processes across interfaces require that bulk phonon modes on both sides have the same frequency. Therefore, for the Si-Ge interface, phonons with frequency above ~10 THz cannot contribute to TBC via elastic processes while they can contribute to TBC via inelastic processes. Inelastic processes allow phonons with different frequencies to exchange energy and contribute to TBC.⁵⁻⁸ The role of the interfacial modes localized at the interfaces is to act as a bridge that interacts with other phonon modes on both sides, which facilitates the inelastic transport across the interfaces for phonons with frequencies higher than 10 THz. This has been theorized in the literature.^{2,3} We note that while we still call this an inelastic process because energy is transferred from one frequency to another across the interface, the underlying mechanism is not originated from a conventional phonon scattering picture. Instead, it is due to the correlation between the localized interfacial phonons and other phonons that extend into the materials making up the interface. In other words, localized interfacial phonons work as a bridge.^{2,3} It is notable that the MD simulations with NNP in this work intrinsically include both elastic and inelastic processes. Other theoretical models such as DMM, AMM, and AGF typically only consider the elastic phonon transport even though the inelastic processes can contribute to the total TBC.⁸⁻¹⁰ The recent development of AGF enables the consideration of both elastic and inelastic contributions but it is very computationally-expensive.¹¹

Revisions: We added the related discussions to Pages 20-21 in the main text.

Comment (6): In fig. 2(b), the dotted vertical lines and the dashed rectangle make it difficult to clearly see the (already weak) intensity around 11-12.5 THz in the interfacial region. Moreover, I am able to see similar intensities in the same frequency region away from the interface on the Si side (particularly, between “-2” and “-3” on the “Distance to interface (nm) axis). Can the authors comment on the origin of these intensities as well?

Response: We appreciate this suggestion. We deleted the dotted lines and dashed rectangle in the figure to avoid any misunderstanding. Those similar intensities in the energy range of 11-12.5 THz on the Si side arise from the bulk phonon modes in the Si. According to the simulated phonon density of states in Fig. 3b, there is a broad peak in the energy range of 12-13.5 THz in the pure Si region. Those modes stem from the optical and acoustic phonon modes near the Brillouin zone (BZ) boundary, according to the phonon dispersion curve. Thus, although those bulk modes locate in the similar frequency to the interfacial phonon modes, they should have very different physical origins and slightly different energy positions. We can use two results to distinguish these two signals. First, in the Supplementary Fig. 4, the interfacial vibrational spectrum has extra intensity near 12.0 THz, compared to the linear combination of bulk Ge and Si vibrational spectra, indicating the existence of additional phonon modes at the interface. Second, we further performed angle-resolved vibrational EELS method to collect the vibrational signals at the BZ center (Γ point, Supplementary Fig. 11a) to exclude the interference of phonon modes near the BZ boundary. Using a combination of a small convergence semi-angle (3 mrad) and a small EELS collection angle, we can obtain the angle-resolved vibrational signal with high momentum resolution (0.5 \AA^{-1}) and appropriate spatial resolution (2.6 nm).¹² Supplementary Fig. 11b depicts local angle-resolved vibrational spectra acquired at the bulk Si, bulk Ge, and interface. Si and Ge spectra contain one major peak at 15.1 THz (62.4 meV) and 8.9 THz (36.8 meV), respectively, which are consistent with the optical phonon modes at the BZ center of Si and Ge.^{13,14} We observed a clear peak at ~ 12.0 THz in the interfacial vibrational spectra while that from the bulk Si at this frequency disappeared. The clear phonon modes at ~ 12.0 THz at the interface without the interference of the Si phonon modes match well with our simulation results.

Supplementary Fig. 11 Momentum resolved vibrational spectra in the Si-Ge heterointerface with a convergence semi-angle of 3 mrad. **a** Reciprocal space diagram including the convergent beam diffraction pattern (CBED) of Si/Ge along [011] direction and the BZs (cyan contours). The center black disk and surrounding gray ones are the transmitted beam and diffracted beams with a radius of 3 mrad. The red circle represents the EELS entrance aperture, which positions at the BZ center to collect the corresponding phonon signals using this angle-resolved condition. The locations of Γ , X, and L points are indicated. **b** Local angle-resolved vibrational spectra of Si, Ge, and interface. Three dashed horizontal lines are the zero baseline.

Revisions: We deleted the dashed lines and the dashed square. We added Supplementary Fig. 11 and related discussions to Pages 18-19 in the Supplementary Information. The discussions about distinguishing the two signals were added to Page 10 in the main text.

Reviewer #2:

The paper is devoted to the experimental Raman and EELS-STEM studies of the vibrational (phonon) modes confined at the interface between Ge and Si lattices. The paper reports the first experimental observation of localized interfacial phonon modes at ~12 THz at a high-quality epitaxial Si-Ge interface. These modes are further confirmed by the molecular dynamics simulations of the thermal boundary conductance through such interface, which is measured in

time-domain thermoreflectance experiments. The paper presents interesting and comprehensive results, confirming by different means the existence of the phonon modes confined at the Si-Ge interface. In my opinion, the paper can be published in Nature Communications with the proper discussion of the physical origin of the observed interface phonon modes.

Response: We thank the reviewer for the positive evaluation of our manuscript as “interesting and comprehensive results” and “can be published in Nature Communications with the proper discussion of the physical origin of the observed interface phonon modes”.

Comment (7): But the physical origin of such modes is not revealed and discussed, which lowers the merit of the paper. The observed interface modes, which are out of the phonon band in Ge crystal, can be related with the vibrations of the light Si atoms that are encapsulated in the effective cages by the surrounding heavy Ge atoms. Such atomic configurations are present in the interfacial mixing domain shown in Fig. 3 and Supplementary Figs. 6 and 7. On the other hand, the resonance phonon mode and transmission at ~13 THz (within the Tersoff interatomic potential) was revealed, e.g., in AIP Conference Proceedings 2241, 020023 (2020), doi: 10.1063/5.0011386, in which phonon transmission through the monolayers of Si atoms, symmetrically encapsulated by the layers of Ge atoms in Si lattice, was studied and different effective cages for light Si atoms were described.

Response: We appreciate the helpful comments and suggestions from the reviewer. We added additional discussions about the physical origin of the interfacial modes in two aspects: first, we visualize the eigenmodes at ~12.4 THz and show that they are mainly localized in the interfacial region that involve Si-Ge bonds (Fig. 5d). Second, we explain that the unique Si-Ge bonds at the interface lead to such modes.

Fig. 5 **d** Eigenvectors for an example interfacial mode at ~ 12.4 THz for a sharp interface and an interface with interfacial mixing. The upper side is Si and the lower side is Ge.

We agree with the reviewer that Si atoms encapsulated by surrounding Ge atoms can lead to modes at a frequency that does not exist in bulk Si and Ge, and the observed frequency is close to that of the interfacial modes.¹⁵ But this encapsulated atomic configuration is not a necessary condition for observing interfacial modes. In our work, we studied both an ideally sharp interface and an interface with a random mixture of Si and Ge atoms. As shown in Supplementary Fig. 7, we observed obvious peaks in the phonon density of states around 12 THz for the sharp Si-Ge interface as well. The Si atoms at the sharp interface have Si-Ge bonds but are not encapsulated by surrounding Ge atoms. The origin of the interfacial modes is thus believed to be from the Si-Ge bonds that do not exist in bulk Si or bulk Ge. Therefore, the vibrations at ~ 12 THz stem from the Si-Ge bonds where the two atoms tend to vibrate differently due to the mass difference compared to the atoms in bulk Si or bulk Ge. We calculate the vibrational modes by finite difference method, and Fig. 5d shows the eigenvectors of an example interfacial mode at ~ 12.4 THz for a sharp interface and an interface with atomic mixing. It is found that the eigenvectors are limited to the two atomic layers at the sharp interface (left panel in Fig. 5d). When the Si-Ge mixture is introduced at the interface, there are more Si-Ge bonds. Si encapsulated by Ge cages is one of such cases. In these cases, interfacial modes exist on atoms in the whole mixed region, as depicted by

the right panel in Fig. 5d. Therefore, for the predominant origin of the interfacial modes at the Si-Ge interface, we believe that the unique Si-Ge bonds at the interface result in the interfacial modes.

Revisions: We added discussions about the reference the reviewer mentioned into the main text (Page 21) and Supplementary Information (Pages 11-12). We also added Fig. 5d to the main text. In addition, we added the discussions about the physical origin of the interfacial modes to Page 21 in the main text.

Reviewer #3:

The authors, for the first time, confirm the existence of interfacial phonon modes by creating high-quality Si-Ge interfaces, using molecular beam epitaxy and detecting the localized modes by Raman spectroscopy and high-energy-resolution electron energy-loss spectroscopy in a scanning transmission electron microscope. These modes are further confirmed using molecular dynamics simulations with a high-fidelity neural network interatomic potential, which also yield TBC agreeing well with that measured from time-domain thermoreflectance (TDTR) experiments. It is an interesting work, however, the present is in poorly written and lacks in-depth analysis. Besides, there are several logical problems existing in the work, and some of anomalous experimental data need to be confirmed. Therefore, a significant modification should be done for current manuscript for further consideration of publication in Nature Communications.

Response: We thank the reviewer for the positive evaluation of our manuscript as “interesting”.

Comment (8): As mentioned in the “Introduction” section, “For insulator or semiconductor related interfaces, lattice vibrations (phonons) are the dominant heat carriers for interfacial thermal transport “, further acoustic phonons rather than optical typically play the predominated role in heat transfer based on the Debye-Callaway model. Nevertheless, peaks indexed to interfacial modes are located at ~11.4-12.8 THz for both Si and Ge atoms shown in Fig. 3, and “The PDOS peaks of Si and Ge optical phonons are ~15 THz and ~9 THz”, thus these additional phonons should belong to optical frequency branch, and would not affect the thermal properties evidently. Therefore, please explain why the interfacial modes “can have non-trivial contribution to TBC”?

Response: We thank the reviewer for this comment. This is a good question. We will explain this in several aspects. First, Debye-Callaway model is based on the phonon gas model assuming perfect periodic lattice matrix, which typically does not hold for interfacial thermal transport due to the symmetry breaking at interfaces. Therefore, thermal conduction in perfect single crystals is very different from that across interfaces. Second, the impression that optical phonons do not affect thermal properties is only true for the thermal conductivity of some bulk single crystals with simple crystal structures. Optical phonons can play an important role in thermal conductivity of nanostructures and materials with complex crystal structures.¹⁶ For example, the optical phonons in β -Ga₂O₃ can contribute to up to 44% of its thermal conductivity.¹⁷ Third, in terms of interfacial thermal transport, it is well-documented that optical phonons can contribute to TBC significantly via inelastic processes.^{5-8,10,11,18} Elastic processes across interfaces require the phonons on both sides have the same frequency, thus the phonons above 10 THz for our Si-Ge interface cannot contribute to TBC through elastic processes. However, inelastic processes allow phonons with different frequencies to exchange energy and contribute to TBC. We note that the interfacial modes, which have a high frequency of ~12 THz, are originated from the Si-Ge bonds unique to the interfacial region. The vibration of this bond is an intra-unit cell phenomenon, so it is similar to the origin of optical phonons. The role of such interfacial modes, which are mainly localized around the interface (Fig. 5d), is to act as a bridge coupling the bulk phonon modes on both sides, which facilitates inelastic processes across interfaces. This has been theorized in the literature.^{2,3,8} We note that while we still call this an inelastic process because energy is transferred from one frequency to another across the interface, the underlying mechanism is not originated from a conventional phonon scattering picture. Instead, it is due to the correlation between the mainly localized interfacial phonons and other phonons that extend into the materials making up the interface. In other words, localized interfacial phonons work as a bridge.^{2,3} It is notable that the MD simulations with NNP in our work intrinsically include both elastic and inelastic contributions to TBC while other theoretical models such as AMM, DMM, and AGF typically cannot include inelastic contribution.^{8,9,19}

Revisions: We added related discussions to Pages 20-21 in the main text to make the above arguments clear.

Comment (9): On page 9, “Clear peaks of the vibrational spectral intensity are observed around the interface at 11.6 and 12.0 THz”, however, the peaks are not clear in Fig. 2d, especially for that of 12.0 THz. Therefore, the evidence based on EELS-STEM is not sufficient, please provide more analysis here and take a deep insight. Otherwise, the existence of interfacial modes should be questioned to some extent.

Response: This comment is very helpful. The major obstacle for the identification of the interfacial modes in the line-scan dataset in Fig. 2 is the bulk phonon modes in the similar frequency of 12-13.5 THz which originate from the optical phonon band near the Brillouin zone (BZ) boundary of Si. To overcome this issue, we collected angle-resolved vibrational EELS at BZ center (Γ point) to exclude the phonon band near the BZ boundary. Supplementary Fig.11 shows the experimental setup and angle-resolved vibrational spectra at Si, Ge, and interface. Only the interfacial vibrational spectrum contains the additional signal at \sim 12.0 THz, confirming the existence of the interfacial phonon modes in that energy range. Additionally, in the Supplementary Fig. 4, the interfacial vibrational spectrum has extra intensity near 12.0 THz, compared to the linear combination of bulk Ge and Si vibrational spectra, indicating the existence of additional phonon modes at the interface.

Revisions: We added Supplementary Fig. 11 and related discussions to Pages 18-19 in the Supplementary Information. The discussions about distinguishing the two signals were also added to Page 10 in the main text.

Comment (10): Is there any experimental evidence of TEM or STEM for the validity of atomic mixing effects shown in Fig. 3(a)? After all, no such difference can be found in the HAADF-STEM image, and sharp interfaces are typically used by previous studies.

Response: We appreciate this constructive comment from the reviewer. Previous theoretical calculations only consider the ideal Si-Ge interfaces, but realistic interfaces cannot be perfect. The intermixing during epitaxial growth of Si and Ge is typically un-avoidable due to the high temperature and strain.^{20,21} Decreasing growth temperature can reduce the atomic diffusion

responsible for interfacial intermixing.²² That is why we grew the samples at a relatively low temperature (400 °C), but it is still un-avoidable to have limited intermixing. To examine the intermixing between Si and Ge, we conducted atomic-resolution STEM imaging and energy-dispersive X-ray spectroscopy (EDS) mapping. Supplementary Fig. 9 shows the atomic-resolution HAADF-STEM images at the Si-Ge interfaces and intensity line profiles across the interface. Since the HAADF signal monotonically increases with the atomic number (Z contrast) of the observed materials, we can use HAADF intensity to distinguish Si and Ge. Thus, the darker and brighter regions in Supplementary Fig. 9a are attributed to Si and Ge, respectively. Supplementary Fig. 10a-c show STEM-EDS results acquired at the interface. Using the line profile of both Si and Ge proportions in Supplementary Fig. 10d, we observed interfacial intermixing as well. We also note that interfacial phonon modes with frequencies around 12 THz exist in both sharp and intermixed Si-Ge interfaces (see Fig. 5d and newly added discussion about the origin of the interfacial modes in the main text). By adding the intermixing in our MD model, we were able to obtain TBC that matches better (than an ideally sharp interface) with the experimental data which were from Si-Ge interface with intermixing. Thus, our experiments and simulations are consistent, and they corroborate the existence of the interfacial modes.

Supplementary Fig. 9 Interfacial structure. **a** Atomic resolution HAADF-STEM image acquired at the Si-Ge interface of the EELS and Raman sample. Scale bar is 2 nm. **b** Line profile of HAADF intensity along the arrow in **a**.

Supplementary Fig. 10 STEM-EDS mapping of the Si-Ge interface of the Raman and EELS sample. **a** HAADF-STEM image acquired at the Si-Ge interface. **b, c** EDS elemental mapping of Si (**b**, red) and Ge (**c**, green) from the same region in **a**. Scale bars in (**a-c**) are all 2 nm. **d** Line profiles of atomic percentage of Si and Ge along the arrow in **a**.

Revisions: We added Supplementary Fig. 9, Supplementary Fig. 10, and related discussions to Pages 16-17 in the Supplementary Information. We have also added discussion about the origin of interfacial modes on Page 21 of the main text.

Comment (11): How to calculate the TBC? The details shown in “Methods” section are confusing and should be clear for this calculation process.

Response: We thank the reviewer for this question. The details about the TBC calculation processes have been described in the original Supplementary Information. To avoid confusion, we moved the details about TBC calculations in the Supplementary Information to the “Methods” section. We hope this will resolve any confusion.

Revisions: We moved the details about TBC calculations from Supplementary Information to the Methods section in the main text (Pages 26-27).

Comment (12): Additional contents should be added. On page 6, “Sample 2 is a 250-nm Ge layer grown on a Si substrate and is used for TBC measurements, where the Ge layer thickness is

optimized to maximize the sensitivity of the TBC of the Si-Ge interface in TDTR measurement.”
The optimization process of Ge layer thickness and TDTR measurements require intuitive data.

Response: This comment is very helpful. We added the details of sensitivity optimization process in the Supplementary Information. The TDTR sensitivities of the Ge-Si TBC and the Ge thermal conductivity as a function of Ge thickness are shown in the figure below. As we can see, a Ge thickness of 250 nm gives us the highest sensitivity of Ge-Si TBC and the lowest sensitivity of Ge thermal conductivity. This is why we chose to grow 250 nm Ge on Si to perform TDTR measurements.

Supplementary Fig.12 TDTR sensitivity as a function of Ge thickness. a Sensitivity of the Ge-Si TBC as a function of Ge thickness. **b** Sensitivity of the Ge thermal conductivity as a function of Ge thickness.

Revisions: We added Supplementary Fig. 12 and related discussions to Page 19 in the Supplementary Information.

Comment (13): On page 18, “The two samples in this work were grown by molecular beam epitaxy (MBE). The silicon substrates (boron doped $5\text{\AA}\sim 1014\text{-}1\text{\AA}\sim 1015$ atoms/cm³) were given a modified RCA cleaning procedure followed by an HF dip as the last step”, please provide the full name of “RCA”.

Response: We appreciate the reviewer for the suggestion. The full name of “RCA” cleaning procedure is Radio Corporation of America (RCA) cleaning procedure. RCA cleaning procedure is a standard set of wafer cleaning steps which need to be performed before high-temperature processing steps of Si wafers.

Revisions: We added the full name of RCA. We also added more details about RCA cleaning in the Supplementary Information (Page 2).

Comment (14): There are many grammar errors existing in this work, the paper should be thoroughly polished. 1) On page 3, “Simulations find that the interfacial phonon modes have obvious contribution to the total TBC”, “an” should be added before “obvious”. 2) On page 4, “Interfaces impede heat flow in micro/nanostructured systems, which highlight the growing importance of thermal management of microelectronics and devices for energy conversion and storage”, “highlight” should be corrected as “highlights”. 3) On page 16, “highlighting the important of correctly capturing the microscopic interface conditions”, “important” should be modified as “importance”; “The fact that NNP-MD is accurate in predicting TBC also give confidence on its predicted non-trivial contribution”, “give” should be replaced with “gives”. 4) Lots of confusing sentences should be optimized, e.g., on page 5, “several molecular dynamics (MD) studies have shown that phonon modes that are unique to the interface (i.e., interfacial phonon modes) can play important roles in interfacial thermal transport”; “these MD simulations were either based on empirical interatomic potentials that were not specifically developed for interfaces or employed approximation for interfacial interactions despite the use of first-principles force constants”...

Response: We appreciate the reviewer for the check of grammar errors and constructive comments to improve our manuscript. We polished the manuscript and revised the grammar errors.

Revisions: We polished the manuscript and double-checked the grammar errors.

REFERENCES

- 1 Chalopin, Y. & Volz, S. A microscopic formulation of the phonon transmission at the nanoscale. *Appl. Phys. Lett.* **103**, 051602 (2013).
- 2 Gordiz, K. & Henry, A. Phonon transport at crystalline Si/Ge interfaces: the role of interfacial modes of vibration. *Sci. Rep.* **6**, 23139 (2016).
- 3 Rohskopf, A., Li, R., Luo, T. & Henry, A. A computational method for studying vibrational mode dynamics. *arXiv preprint arXiv:2108.04795* (2021).
- 4 Glassbrenner, C. J. & Slack, G. A. Thermal conductivity of silicon and germanium from 3 K to the melting point. *Phys. Rev.* **134**, A1058 (1964).
- 5 Hopkins, P. E., Norris, P. M. & Stevens, R. J. Influence of inelastic scattering at metal-dielectric interfaces. *J. of Heat Transf.* **130**, 022401 (2008).
- 6 Hopkins, P. E. Multiple phonon processes contributing to inelastic scattering during thermal boundary conductance at solid interfaces. *J. of Appl. Phys.* **106**, 013528 (2009).
- 7 Murakami, T., Hori, T., Shiga, T. & Shiomi, J. Probing and tuning inelastic phonon conductance across finite-thickness interface. *Appl. Phys. Exp.* **7**, 121801 (2014).
- 8 Feng, T., Zhong, Y., Shi, J. & Ruan, X. Unexpected high inelastic phonon transport across solid-solid interface: Modal nonequilibrium molecular dynamics simulations and Landauer analysis. *Phys. Rev. B* **99**, 045301 (2019).
- 9 Cheng, Z. *et al.* Thermal conductance across harmonic-matched epitaxial Al-sapphire heterointerfaces. *Commun. Phys.* **3**, 1-8 (2020).
- 10 Lyeo, H.-K. & Cahill, D. G. Thermal conductance of interfaces between highly dissimilar materials. *Phys. Rev. B* **73**, 144301 (2006).
- 11 Dai, J. & Tian, Z. Rigorous formalism of anharmonic atomistic Green's function for three-dimensional interfaces. *Phys. Rev. B* **101**, 041301 (2020).
- 12 Yan, X. *et al.* Single-defect phonons imaged by electron microscopy. *Nat.* **589**, 65-69 (2021).
- 13 Wei, S. & Chou, M. Phonon dispersions of silicon and germanium from first-principles calculations. *Phys. Rev. B* **50**, 2221 (1994).
- 14 Aouissi, M., Hamdi, I., Meskini, N. & Qteish, A. Phonon spectra of diamond, Si, Ge, and α -Sn: Calculations with real-space interatomic force constants. *Phys. Rev. B* **74**, 054302 (2006).

- 15 Kosevich, Y. A. & Strelnikov, I. Extraordinary phonon transmission through hidden lattice-wave nanochannels as resonance quantum phonon tunneling, *AIP Conf. Proceed.* **2241**, 1, 020023, AIP Publishing LLC, (2020).
- 16 Tian, Z., Esfarjani, K., Shiomi, J., Henry, A. S. & Chen, G. On the importance of optical phonons to thermal conductivity in nanostructures. *Appl. Phys. Lett.* **99**, 053122 (2011).
- 17 Santia, M. D., Tandon, N. & Albrecht, J. Lattice thermal conductivity in β -Ga₂O₃ from first principles. *Appl. Phys. Lett.* **107**, 041907 (2015).
- 18 Gaskins, J. T. *et al.* Thermal boundary conductance across heteroepitaxial ZnO/GaN interfaces: assessment of the phonon gas model. *Nano Lett.* **18**, 7469-7477 (2018).
- 19 Giri, A. & Hopkins, P. E. A review of experimental and computational advances in thermal boundary conductance and nanoscale thermal transport across solid interfaces. *Adv. Funct. Mater.*, 1903857 (2019).
- 20 Chen, P. *et al.* Role of surface-segregation-driven intermixing on the thermal transport through planar Si/Ge superlattices. *Phys. Rev. Lett.* **111**, 115901 (2013).
- 21 Chaparro, S. *et al.* Strain-driven alloying in Ge/Si (100) coherent islands. *Phys. Rev. Lett.* **83**, 1199 (1999).
- 22 Van Nostrand, J. E., Chey, S. J., Hasan, M.-A., Cahill, D. G. & Greene, J. Surface morphology during multilayer epitaxial growth of Ge (001). *Phys. Rev. Lett.* **74**, 1127 (1995).

REVIEWERS' COMMENTS

Reviewer #1 (Remarks to the Author):

The authors have addressed all of my comments and concerns satisfactorily. I recommend the publication of the manuscript in its current form in Nature Communications.

Reviewer #2 (Remarks to the Author):

Manuscript NCOMMS-21-25101-T

"Experimental Observation of Localized Interfacial Phonon Modes"
by Zhe Cheng et al.

Second Referee Report

The revised paper presents further arguments that support the authors claim on the observation of localized interfacial phonon modes. In particular, it is emphasized that the modes are present even at the sharp Ge-Si interface, which is illustrated in Fig. 5d and in Supplementary Fig. 7. The authors believe that the unique Si-Ge bonds at the interface result in the interfacial modes.

This is plausible, but then the natural question arises how the phonon mode with the frequency of ~ 12.4 THz propagates through the layer of Ge of 250 nm thickness, shown in Supplementary Fig. 2a, in the band gap frequency region of this material that starts at ~ 10 THz? Such band-gap phonon propagation, which contributes to the thermal boundary conductance in the considered system, can occur only due to inelastic (nonlinear) phonon processes. Again, the concrete physical mechanism for such inelastic phonon transmission would be desirable to mention in the paper. One of the possible nonlinear mechanism can be related with the temperature-dependent subharmonic and higher-harmonic phonon transmission provided by anharmonic atomic vibrations in the local interface mode that was discussed, e.g., in PRB 52, 1017 (1995) for vibrationally mismatched interfaces.

In my opinion, the paper can be published in Nature Communications with the addition of the discussion of the possible physical mechanism of the inelastic phonon transmission in the band-gap region through the Ge layer.

Reviewer #3 (Remarks to the Author):

The authors have addressed all my questions, I suggest it can be published in Nat. Commun.

We sincerely thank the reviewers and the editor for the precious time and attention on our manuscript. We also greatly appreciate the constructive and valuable comments from the reviewers. These insightful suggestions and corresponding revisions improve our manuscript and make it more complete.

Reviewer #2:

The revised paper presents further arguments that support the authors' claim on the observation of localized interfacial phonon modes. In particular, it is emphasized that the modes are present even at the sharp Ge-Si interface, which is illustrated in Fig. 5d and in Supplementary Fig. 7. The authors believe that the unique Si-Ge bonds at the interface result in the interfacial modes.

This is plausible, but then the natural question arises how the phonon mode with the frequency of ~ 12.4 THz propagates through the layer of Ge of 250 nm thickness, shown in Supplementary Fig. 2a, in the band gap frequency region of this material that starts at ~ 10 THz? Such band-gap phonon propagation, which contributes to the thermal boundary conductance in the considered system, can occur only due to inelastic (nonlinear) phonon processes. Again, the concrete physical mechanism for such inelastic phonon transmission would be desirable to mention in the paper. One of the possible nonlinear mechanism can be related with the temperature-dependent subharmonic and higher-harmonic phonon transmission provided by anharmonic atomic vibrations in the local interface mode that was discussed, e.g., in PRB 52, 1017 (1995) for vibrationally mismatched interfaces.

In my opinion, the paper can be published in Nature Communications with the addition of the discussion of the possible physical mechanism of the inelastic phonon transmission in the band-gap region through the Ge layer.

Response: We thank the reviewer for the positive feedback and directing us to the reference above. As for the question about inelastic phonon processes, there have been simulation studies showing that localized interfacial modes actually act as a bridge coupling the bulk phonon modes on both sides, which facilitates the inelastic transport across the interfaces for phonons with different

frequencies. Such contributions are due to the correlation between the localized interfacial phonons and other phonons that extend into the materials making up the interface. It is notable that the MD simulations with NNP in this work intrinsically include both elastic and inelastic processes. We have added discussions on page 20 of the revised manuscript and cited the paper together with other papers as the following:

“It is noted that the frequencies of the interfacial modes are higher than those of the optical phonons of bulk Ge. Elastic processes across interfaces require that bulk phonon modes on both sides have the same frequency. Therefore, for the Si-Ge interface, phonons with frequencies above ~ 10 THz cannot contribute to TBC via elastic processes while they can contribute to TBC via inelastic processes. Inelastic processes allow phonons with different frequencies to exchange energy and contribute to TBC.^{18,28-31} Previous simulations have shown that the role of the interfacial modes localized at the interface is to act as a bridge coupling the bulk phonon modes on both sides, which facilitates the inelastic transport across the interfaces for phonons with different frequencies.^{17,28,32} We note that while we still call this an inelastic process because energy is transferred from one frequency to another across the interface, the underlying mechanism is not originated from a conventional phonon scattering picture. Instead, it is due to the correlation between the localized interfacial phonons and other phonons that extend into the materials making up the interface.^{17,32} It is notable that the MD simulations with NNP in this work intrinsically include both elastic and inelastic processes. However, other theoretical models such as DMM, AMM, and AGF typically only consider the elastic phonon transport even though the inelastic processes can contribute to the total TBC.^{3,28,33} The recent development of AGF enables the consideration of both elastic and inelastic contributions but it is very computationally expensive.¹³”